# LC-MS/MS metabolomics-facilitated identification of the active compounds responsible for anti-allergic activity of the ethanol extract of *Xenostegia tridentata*

**Rinrada Suntivich**[1], **Worawat Songjang**[2,3]°, **Arunya Jiraviriyakul**[2,3]°,
**Somsak Ruchirawat**[4,5,6], **Jaruwan Chatwichien**[4,6]*

1 National Center for Genetic Engineering and Biotechnology (BIOTEC), National Science and Technology Development Agency (NSTDA), Pathum Thani, Thailand, 2 Department of Medical Technology, Faculty of Allied Health Sciences, Naresuan University, Phitsanulok, Thailand, 3 Integrative Biomedical Research Unit (IBRU), Faculty of Allied Health Sciences, Naresuan University, Phitsanulok, Thailand, 4 Program in Chemical Sciences, Chulabhorn Graduate Institute, Chulabhorn Royal Academy, Bangkok, Thailand, 5 Laboratory of Medicinal Chemistry, Chulabhorn Research Institute, Bangkok, Thailand, 6 Center of Excellence on Environmental Health and Toxicology (EHT), OPS, Ministry of Higher Education, Science, Research and Innovation, Bangkok, Thailand

☯ These authors contributed equally to this work.
* jaruwanc@cgi.ac.th

**Data Availability Statement:** All relevant data are within the manuscript and its Supporting information files.

## Abstract

*In vivo* and *in vitro* anti-allergic activities of ethanol extract of *Xenostegia tridentata* (L.) D.F. Austin & Staples were investigated using passive cutaneous anaphylaxis reaction assay and RBL-2H3 cell degranulation assay, respectively. The crude ethanol extract exhibited promising activities when compared with the known anti-allergic agents, namely dexamethasone and ketotifen fumarate. The ethyl acetate subfraction showed the highest anti-allergic activity among various sub-partitions and showed better activity than the crude extract, consistent with the high abundance of total phenolic and flavonoid contents in this subfraction. LC-MS/MS metabolomics analysis and bioassay-guided isolation were then used to identify chemical constituents responsible for the anti-allergic activity. The results showed that major components of the ethyl acetate subfraction consist of 3,5-dicaffeoylquinic acid, quercetin-3-O-rhamnoside, kaempferol-3-O-rhamnoside and luteolin-7-O-glucoside. The inhibitory activity of the isolated compounds against mast cell degranulation was validated, ensuring their important roles in the anti-allergic activity of the plant. Notably, besides showing the anti-allergic activity of *X. tridentata*, this work highlights the role of metabolomic analysis in identifying and selectively isolating active metabolites from plants.

## Introduction

Allergic diseases including asthma, rhinitis, atopic dermatitis and food allergies are common chronic health problems worldwide [1], possibly leading to life-threatening conditions.

**Funding:** This work was supported by the Agricultural Research Development Agency (Public Organization) [grant numbers CRP6305030720] and the National Science and Technology Development Agency [grant number FDA-CO-2563-12800-TH]. The funders had no role in study design, data collection and analysis, decision to publish, or preparation of the manuscript.:JC.

**Competing interests:** The authors have declared that no competing interests exist.

Hypersensitivity reactions can be divided into type I—IV. Type I hypersensitivity is mediated by immunoglobulin E (IgE) and proceeds by a two-step mechanism; the first exposure (sensitization) and a subsequent exposure to a specific allergen, resulting in a process called mast cell degranulation [2–6]. Histamine and other proinflammatory mediators are then released from mast cells, causing symptoms such as a runny nose, coughing, itchiness and red rashes. Currently, there are different kinds of allergy medications with different modes of action such as anti-histamines, leukotriene receptor antagonists, mast-cell stabilizers, and corticosteroids. None of the available medications, however, can completely cure allergies. In addition, patients usually suffer from drug side effects or resistance after prolonged treatment [7]. Novel drugs for the treatment and prevention of allergies are therefore essential.

Secondary metabolites from plants have been shown to prevent or ameliorate allergic disorders [8]. Natural flavonoids and polyphenolic metabolites possess promising anti-allergic and immune-modulating effects [9–11]. Many flavonoids inhibit mast cell chemical mediator release and cytokine production [9, 10, 12]. Flavonoids such as ayanin, apigenin and fisetin were shown to inhibit IL-4 production with $IC_{50}$ values of 2–5 μM [10]. Luteolin and diosmetin are two of the most potent anti-allergic flavonoids [13]. *In vitro* and *in vivo* studies have shown anti-allergic activity of luteolin and diosmetin against both the early and late phase reactions by inhibiting mast cell degranulation and IgE-mediated TNF-α and IL-4 productions [12–14]. These findings have suggested the potential of utilizing flavonoid-rich plants for complementary or alternative treatment of allergic diseases.

Metabolomics is a comprehensive measurement of metabolites in biological systems. Metabolomics-based analysis usually aims to identify or quantify small molecules (<1000 Da) by using various analytical techniques such as nuclear magnetic resonance (NMR), gas chromatography mass spectrometry (GC-MS), and liquid chromatography mass spectrometry (LC-MS) [15]. Metabolomics accelerates the discovery of bioactive molecules by prioritizing compounds of interest before isolation steps [16–18]. LC/MS-based metabolomics approach has proven to be an efficient tool for studying diverse plant metabolites [19–21], which are estimated to be more than 200,000 metabolites [22]. Identification of compounds of interest can be done by first using the accurate mass obtained from an MS experiment to predict a possible chemical formula. Once the chemical formula is known, fragmentation information of MS/MS obtained from collision-induced dissociation (CID) and/or high collision dissociation (HCD) could guide compound identification [23, 24]. Compound annotation by matching MS/MS data to the database and integrating it with statistical analysis [25] can increase the level of confidence for identifying potential active compounds. Metabolomics could therefore serve as a powerful tool in plant-derived natural product research [20].

*Xenostegia tridentata* (L.) D.F. Austin & Staples (also known as *Merremia tridentata* (L.) Hallier f. or "Thao tot ma" in Thailand) is a medicinal plant widely distributed in tropical regions such as Africa, India, China, Australia and Southeast Asia [26–28]. This plant is a perennial herb grown on disturbed sites such as roadsides, grasslands and cultivated areas. The plant has been reported as a component in Ayurvedic medicine for many diseases, including rheumatism, skin infections, fever, diabetes, diarrhea and urinary disorders [29, 30]. Studies have shown that *X. tridentata* possesses broad biological activities including anti-oxidant, anti-diabetic, anti-inflammatory, anti-arthritis, analgesic, wound healing, anti-microbial and larvicidal activities [31–38]. These activities are attributed presumably to flavonoids, major phytochemical components present in this plant. *X. tridentata* extracts which are considered safe and have no observed toxicity in rats when orally administered at a concentration of up to 2000 mg/kg [31, 32, 34, 38].

Despite the availability of numerous studies on phytochemicals and bioactivities of *X. tridentata*, knowledge of the chemical structures of *X. tridentata*-derived compounds is limited

[11, 26, 31, 39, 40]. Up to date, there have been only four flavonoids, including diosmetin, luteolin, diosmetin-7-O-β-D-glucoside and luteolin-7-O-β-D-glucoside isolated from this plant [11]. Moreover, in-depth identification of chemical constituents responsible for the bio-activities of this plant has never been revealed. Previous reports on the presence of luteolin and diosmetin and the anti-oxidant, anti-inflammatory and anti-arthritis properties that are closely related to anti-allergic activities prompted us to investigate the anti-allergic activity of *X. tridentata*, a never before investigated property. Herein, we studied the *in vivo* and *in vitro* inhibitory effects of the ethanol extract of *X. tridentata* on type I hypersensitivity. To identify the active components responsible for the activity, the crude ethanol extract was further fractionated and the resulting subfractions were evaluated for inhibitory activity on IgE-mediated degranulation of RBL-2H3 cells. Based on the assumption that fractions with high inhibitory activity contain high amount of the active compounds, LC-MS/MS metabolomics analysis to identify and quantify metabolites in each fraction was performed. The features that were highly abundant in the potent fraction, relative to the others, were listed, and their chemical structures and anti-allergic activities were validated.

## Materials and methods

### General information

All solvents and chemicals used were at least of analytical grade and were used without further purification. Folin & Ciocalteu′s phenol reagent and reference compounds (gallic acid, quercetin, luteolin and kaempferol) were purchased from Sigma Aldrich. Solvents were purchased from RCI Labscan Limited. Reagents for cell culture experiments including cell culture media, fetal bovine serum, penicillin-streptomycin, 3-(4,5-dimethylthiazol-2-yl)-2,5-diphenyltetrazolium bromide (MTT) and 2,4-dinitrophenylated bovine serum albumins (DNP-BSA) were purchased from Thermo Fisher Scientific (Thailand) Co., Ltd.

### Plant material

The aerial part of *X. tridentata* was collected from Chonburi province, Thailand. The specimen was deposited at the PNU plant herbarium, Department of Biology, Faculty of Science, Naresuan University, Thailand (Voucher ID: 004662). The sample was rinsed with tap water and shade dried for 3 days before grinding to powder. The powder was kept at -20˚C until used.

### Preparation of *X. tridentata* extracts

Two hundred grams of *X. tridentata* powder was macerated in 2000 mL absolute ethanol (AR grade) for 48 hours. The supernatant was filtered through cotton and subsequently re-filtered through filter paper (Whatman No.1) by vacuum filtration. The clear filtrate was concentrated under vacuum (150 mbar, 40 ˚C) to yield 26.29 g of green oil (called F1: crude EtOH). Crude EtOH extract (24.5 g) was added with 100 mL of distilled water. Subsequent extractions with hexane (8 x 100 mL), ethyl acetate (5 x 100 mL) and *n*-butanol (3 x 100 mL) were performed. The resulting partitions were combined and concentrated under vacuum to yield four more fractions, namely F2: EtOH/Hex (green oil) 11.2 g, F3: EtOH/EtOAc (yellowish brown oil) 0.9 g, F4: EtOH/BuOH (brown oil) 2.6 g and F5: EtOH/H$_2$O (brown oil) 4.6 g.

### Ethics statement

The animal experiment protocol was approved by Naresuan University Animal Care and Use Committee according to protocol number NU-AE630608. The mice were given free access to

standard food and water, with 12-hour light-dark cycle at 22±2˚C. For euthanasia, mice were intraperitoneally administered with sodium thiopental (> 100 mg/kg).

## Passive cutaneous anaphylaxis reaction assay

Experiments were performed on 7-8-week-old adult ICR mice (Siam Nomura, Thailand). The experiments were carried out at the Center for Animal Research, Naresuan University. To induce a passive cutaneous anaphylaxis reaction, ears of ICR mice were intradermally injected with 0.5 μg of mouse anti-dinitrophenyl (DNP) IgE antibodies (Sigma Aldrich) or normal saline solution (NSS). After 24 hours, the filtered crude EtOH extract (F1) or dexamethasone resuspended in 1.2%TWEEN in NSS was orally administered at indicated concentrations. The 1.2%TWEEN in NSS was administered in the vehicle control group. One hour later, mice were intravenously injected with dinitrophenylated human serum albumin (DNP-HSA) 60 μg containing 0.5% Evans blue in phosphate buffer saline (PBS). Mice were observed for another hour and euthanized. Ears were photographed and removed for the Evans blue elution assay. The ears were cut into small pieces and dissolved in 400 μL of formamide at 63 ºC overnight. The resulting supernatant was then collected, and the absorbance was measured at 630 nm using a microplate reader (EnSpire™ Multimode Plate Reader, PerkinElmer, Inc.). Data was presented as absorbance compared to the control group. All efforts were made to minimize suffering and the number of mice needed to reach statistical significance and experimental reproducibility.

## Determination of total phenol and total flavonoid contents

**Total phenolic content.**   Total phenolic compounds content (TPC) was determined by Folin-Ciocalteu colorimetric method which compared absorbance at 765 nm ($A_{765}$) of unknown samples to a standard curve of $A_{765}$ of serial dilutions of gallic acid solution (0–200 μg/mL). The protocol reported by Siriwoharn *et al.* [41]. was followed with some modifications. Briefly, 100 μL of 1:10 diluted Folin-Ciocalteu Phenol reagent was added to 25 μL of gallic acid dilutions or unknown samples. The mixture was incubated in the dark for 5 minutes at ambient temperature, then 60 μL of 10% w/v sodium carbonate was added and incubated in the dark for 30 minutes. Absorbance at 765 nm was measured. Results were reported as mg of gallic acid equivalent (GAE) per g of crude extract. Each analysis of phenolic compounds in the extracts was done in triplicate.

**Total flavonoid content.**   The total flavonoid content was determined by aluminum chloride colorimetric method, which compared the absorbance at 415 nm ($A_{415}$) of unknown samples to a standard curve of $A_{415}$ of serial dilutions of quercetin solution (0–200 μg/mL). The protocol reported by Chang *et al.* [42]. was followed with some modifications. Briefly, 75 μL of 95% ethanol (v/v) was added to 25 μL of quercetin dilutions or unknown samples. Then, 5 μL of 10% $AlCl_3.6H_2O$, 5 μL of 1M sodium acetate and 140 μL of deionized water were added, respectively. The mixtures were incubated for 30 minutes in the dark at ambient temperature. Absorbance at 415 nm was measured. Results were reported as mg of quercetin equivalent (QE) per g of crude extract. The experiment was done in triplicate.

## LC-MS/MS profiling of the extracts of *X. tridentata*

Liquid chromatography—mass spectrometry was performed using Dionex Ultimate™ 3000 RS Ultra—High Performance Liquid Chromatography (UHPLC)—Orbitrap Fusion™ Tribrid™ Mass Spectrometer (Thermo, Massachusetts, USA), equipped with electrospray ionization (ESI) source. The samples were dissolved in MS-grade methanol to make a 1 mg/mL solution, followed by centrifugation at 14,000 x g for 5 minutes to remove insoluble debris.

Subsequently, the methanol-dissolved samples were filtered through a 0.2 μm PVDF membrane filter and transferred to a glass-insert vial for LC injection. The liquid chromatography—mass spectrometry was performed on a BEH C18 column (1.7 μm diameter, 2.1 x 100 mm) maintained at 60˚C. The optimal mobile phase system was 0.0–2.0 min, 5%-25%B; 2.0–5.0 min, 25%-30%B; 5.0–5.5 min, 30%-45%B; 5.5–9.0 min, 45%-95%B; 9.0–12.5 min, 95%-95%B; 12.5–12.6 min, 95%-5%B; 12.6–15.0 min and 5%-5%B where solvent A was water with 0.1% formic acid and solvent B was acetonitrile with 0.1% formic acid. The mass spectrometer was operated in both ESI positive and negative modes. Source-dependent parameters were as follows: sheath gas ($N_2$), 45 arb; auxiliary gas ($N_2$), 13 arb; sweep gas, 1 arb; ion spray voltage, 3.50 kV for ESI positive and 2.5kV for ESI negative; capillary temperature, 342 ˚C; vaporizer temperature, 358˚C. The scan mode was a full MS1 scan followed by a data-dependent MS2 scan with a mass range of m/z 100–1200. The resolution of the full MS1 scan and the data-dependent MS2 scan were 120,000 and 15,000, respectively. For the data-dependent MS2 scan, the S-Lens RF level was set to 60%, and the automatic gain control (AGC) target was set to 1.0 x $10^5$. Each sample was analyzed in triplicate. A pooled sample, combining an equal fraction of all ten samples, was injected every 10–15 samples to ensure system stability and reproducibility. The MS calibration was conducted using Pierce LTQ Velos ESI Positive Ion Calibration (PSP3A 88323) and Pierce ESI Negative Ion Calibration (PSP3A 88324) according to the manufacturer's protocol.

## Active compounds isolation and chemical structure determination

The fraction with the highest anti-allergic activity (F3: EtOH/EtOAc) was further purified to obtain major chemical components for structural identification and activity validation. One gram of the F3: EtOH/EtOAc fraction was purified by using medium-pressure liquid chromatography (MPLC; PuriFlash 450, Interchim, France) over reversed phase silica gel (PF-15C18AQ-F0080) and eluted with 3%-97% MeOH/$H_2O$. The resulting fractions were examined by thin layer chromatography (TLC). The fractions containing identical compounds were combined and concentrated under vacuum. The NMR spectra of the isolated compounds were recorded on a Bruker Avance NMR spectrometer operating at 300 MHz for $^1$H and 75 MHz for $^{13}$C. High-resolution mass spectra were obtained using ESI on an Orbitrap Fusion™ Tribrid™ mass spectrometer (Thermo, Massachusetts, USA).

## Measurement of RBL-2H3 cell viability

Rat basophilic leukemia cell line, RBL-2H3 (ATCC CRL-2256), was purchased from the American Type Culture Collection (ATCC). Cells were cultured in Dulbecco's modified Eagle's medium (DMEM) supplemented with 10% fetal bovine serum, 100 U/mL penicillin and 100 μg/mL streptomycin at 37˚C with 5% $CO_2$.

The RBL-2H3 cells were seeded at $10^4$ cells/well in a 96-well plate and allowed to grow for 24 hours. Plant extracts were added and allowed to incubate with the cells for 48 hours. The cell viability was determined by using the MTT viability assay. Briefly, culture media was removed. MTT solution (0.5 mg MTT in 1 mL of complete medium) was added and allowed to incubate at 37˚C for 3 hours. The solution was then replaced with 100 μL of DMSO. Absorbance at 570 nm was measured using a microplate reader (Varioskan™ LUX multimode microplate reader, Thermo Scientific™). Percent viability was calculated relative to the untreated control.

## β-Hexosaminidase release assay

IgE-mediated mast cell degranulation was determined by measuring the release of β-hexosaminidase enzyme. Briefly, the RBL-2H3 cells were seeded in 24-well plates at $1 \times 10^5$ cells/well and grown for 24 hours. The cells were sensitized with 200 ng/mL DNP-specific IgE (Sigma Aldrich) for 24 hours. The IgE-sensitized RBL-2H3 cells were washed with PBS and incubated with plant extracts or ketotifen fumarate in Siraganian buffer (119 mM NaCl, 5 mM KCl, 5.6 mM glucose, 0.4 mM $MgCl_2$, 25 mM PIPES, 40 mM NaOH, 1 mM $CaCl_2$, and 0.1% BSA, pH 7.2) for 2 hours at 37°C. Subsequently, the cells were stimulated with DNP-BSA (1 μg/mL) for 1 hour at 37°C. To determine the amounts of β-hexosaminidase released into the supernatant, 50 μL of the supernatant was transferred into a 96-well plate and incubated with 50 μL of substrate solution (0.3 mg/mL p-nitrophenyl N-acetyl-D-glucosamine (Sigma Aldrich) in 0.1M citrate buffer (pH 4.5)) for 2 hours. The remaining cells were lysed with 200 μL of 1% (v/v) Triton X-100 in Siraganian buffer. The amount of β-hexosaminidase inside the cells was determined by incubating 50 μL of the cell lysate with the substrate as indicated above. The enzyme reaction was terminated by adding 200 μL of carbonate buffer containing 0.1 M $Na_2CO_3$ and 0.1 M $NaHCO_3$ (pH 10), and absorbance at 405 nm ($A_{405}$) was measured using a microplate reader (Varioskan™ LUX multimode microplate reader, Thermo Scientific™). The experiment was performed in triplicate. The β-hexosaminidase release ratio (% of max) was calculated according to the following equations.

$$\beta\text{-hexosaminidase released ratio} = A_{405}(\text{supernatant})/[A_{405}(\text{supernatant}) + A_{405}(\text{cell lysate})]$$

$$\beta\text{-hexosaminidase release ratio (\% of max)} = [\beta\text{-hexosaminidase released ratio (sample)}/\beta\text{-hexosaminidase released ratio (vehicle control)}] \times 100$$

For compound 48/80-stimulated degranulation, RBL-2H3 cells ($1 \times 10^5$ cells/ well) were grown on 24-well plates for 24 hours before being incubated for 2 hours at 37°C with a solution of tested samples or ketotifen fumarate in Siraganian buffer, having DMSO as a vehicle control. Subsequently, the cells were stimulated with compound 48/80 (10 μg/mL, Sigma Aldrich) for 1 hour at 37°C. The degree of degranulation was then measured in the same manner as in the case of IgE-mediation.

## Measurement of intracellular reactive oxygen species (ROS) level

The intracellular levels of ROS were measured by using 2',7'-dichlorofluorescein diacetate (DCFH-DA), a cell permeable dye that can be hydrolyzed by cellular esterase and rapidly oxidized to florescent 2',7'-dichlorofluorescein (DCF) by intracellular oxidants. Briefly, the RBL-2H3 cells or IgE-sensitized cells ($10^5$ cells/well) in a 96-well plate were incubated with 5 μM DCFH-DA (Sigma Aldrich) for 1 hour at 37°C. The cells were washed twice with Hanks' balanced salt solution (HBSS) and incubated with samples or DMSO in HBSS for 1 hour. After stimulation with compound 48/80 or DNP-BSA, the fluorescence (excitation/emission: 485/527 nm) of oxidized DCF was analyzed at 30-second intervals for 16 minutes using a microplate reader (Varioskan™ LUX multimode microplate reader, Thermo Scientific™).

## Statistical analysis

One-way analysis of variance (ANOVA) with Tukey's multiple-comparison posttest was performed using GraphPad Prism 7.0 Software. All data were presented as mean ± standard error

of the mean (SEM). Differences between groups were considered statistically significant at a P value of 0.05.

## Results and discussion

### *In vivo* effect of the crude ethanol extract of *X. tridentata* on type 1 hypersensitivity reaction

A passive cutaneous anaphylaxis reaction assay was performed to evaluate the ability of *X. tridentata* extract to attenuate type 1 allergic reaction. Mouse ears were sensitized with anti-DNP IgE. After orally administered with the ethanol extract of *X. tridentata* (F1) or dexamethasone (a known immune suppressive agent), mice were intravenously challenged with antigen (DNP-HSA in 1% Evans blue dye) to stimulate a passive cutaneous anaphylaxis reaction, which could be determined from the vascular permeability of Evans blue dye into the ears. In mice treated with the extract (100 mg/kg) or dexamethasone (10 mg/kg), less Evans blue leakage was observed when compared with the vehicle control group (Fig 1), indicating that passive cutaneous anaphylaxis reaction was significantly suppressed by the extract, in the same manner as dexamethasone. The result suggested an anti-allergic potential of the ethanol extract of *X. tridentata*. However, treatment with the extract at 1000 mg/kg showed a less suppressive effect than at 100 mg/kg. The observed result was possibly related to the pharmacokinetics of

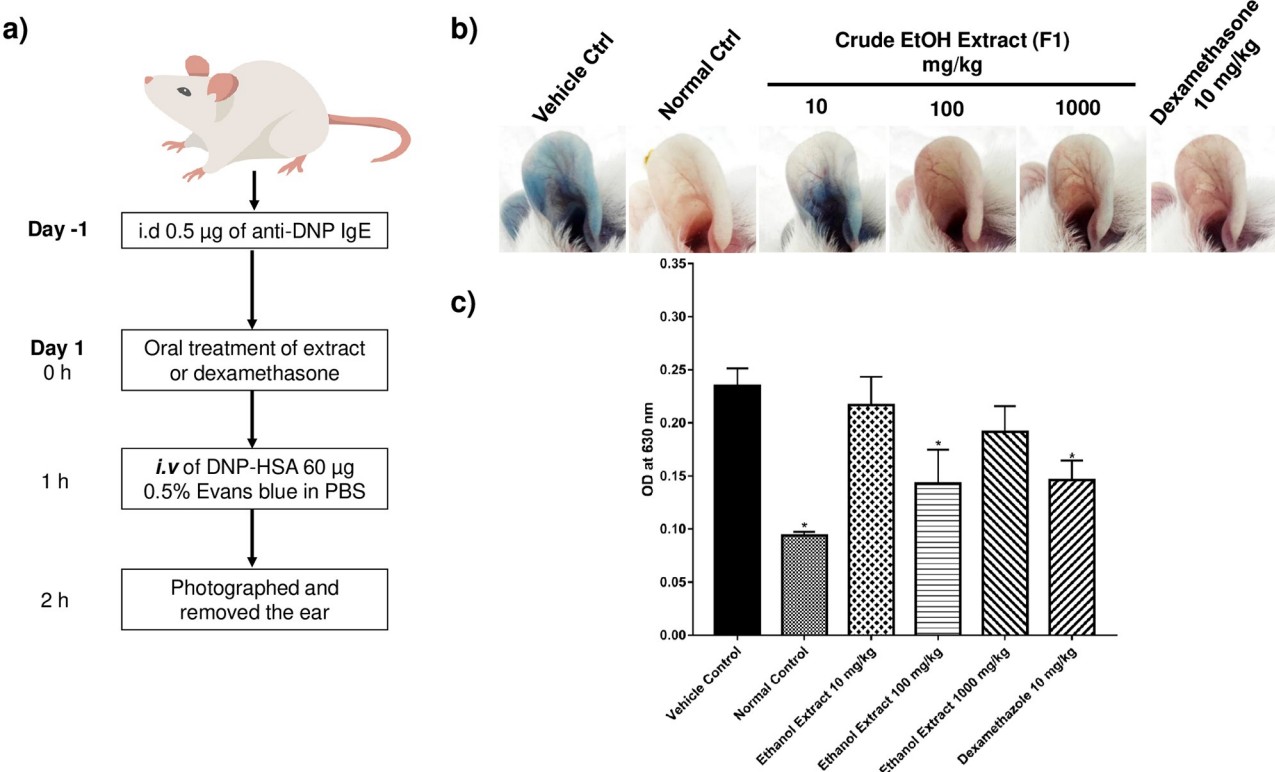

**Fig 1. *In vivo* inhibitory effect of crude EtOH (F1) on IgE-mediated passive cutaneous anaphylaxis reaction.** a) Experimental design of passive cutaneous anaphylaxis assay. Mouse ears were intradermally injected (i.d.) with anti-DNP-IgE. Twenty-four hours post sensitization, mice were orally administrated with F1 or dexamethasone then intravenously (i.v.) challenged with DNP antigen. Evans blue leakage was monitored at 2 hours after stimulation. b) Evans blue leakage after the antigen challenge. Dexamethasone was used as a positive control. c) Evans blue extracted from each ear was quantified using a spectrophotometer. Data are presented as mean ± SEM (n = 5 per group). $^{*}p < 0.05$, compared with the vehicle control group.

the extract. Further study of molecular mechanisms and pharmacokinetics of the extract will therefore be important for optimizing the treatment doses and conditions.

## Inhibitory activity of the extracts on IgE-mediated mast cell degranulation

To determine anti-allergic components in *X. tridentata*, the crude ethanol extract (F1) was dissolved in water and further separated by subsequent extractions with hexane, ethyl acetate and *n*-butanol, yielding samples F2-F5, respectively (Scheme 1). *In vitro* inhibitory activity of the resulting extracts on mast cell degranulation was performed on DNP-specific IgE-sensitized RBL-2H3 cells. The degree of degranulation was determined by the amount of β-hexosaminidase release upon stimulation with the antigen (DNP-BSA). As shown in Table 1, similar to the effect of ketotifen fumarate (a known anti-allergic agent), the F1-F3 extracts possessed mast cell degranulation inhibitory activity on IgE-mediated in a dose-dependent manner. Among them, the ethyl acetate fraction (F3) exhibited the highest anti-allergic activity and showed improved activity over the crude ethanol extract (F1), without being cytotoxic to the cells at the concentration up to 2000 μg/mL. The butanol fraction (F4) showed weak activity at 500 μg/mL and was over 20% toxic to the cells at the higher concentrations. The water fraction (F5) also exerted weak activity and did not inhibit the degranulation in a dose-dependent manner. Markedly, at the concentrations of 1000 and 2000 μg/mL, F3 exhibited significantly higher inhibitory activity than F5: $EtOH/H_2O$ ($p<0.001$). Moreover, to assure that the observed effects of the extracts on the inhibition of mast cell degranulation were not from the inhibition of the enzyme activity, β-hexosaminidase inhibitory activity of each extract was determined. The results showed that no significant inhibition was observed for all tested samples (S1 Table).

**Table 1. Effects of *X. tridentata* extracts on IgE-mediated mast cell degranulation and cell viability of RBL-2H3 cells.**

| Sample | β-hexosaminidase release ratio (% of max) [a] | | | | |
|---|---|---|---|---|---|
| | Treatment concentration (μg/mL) | | | | |
| | *25* | *100* | *500* | *1000* | *2000* |
| F1: crude EtOH | ND | ND | 66.2±8.2 | 34.8±5.5** | N/A |
| F2: EtOH/Hex | ND | ND | 66.4±8.7 | 37.3±6.2** | 28.9±6.5** |
| F3: EtOH/EtOAc | ND | ND | 48.2±2.4** | 28.1±5.9** | 13.8±2.3** |
| F4: EtOH/BuOH | ND | ND | 77.8±2.5 | N/A | N/A |
| F5: EtOH/H₂O | ND | ND | 53.2±2.1** | 63.0±2.1** | 68.0±4.8 |
| ketotifen fumarate | 70.8±4.4* | 37.6±4.9** | ND | ND | ND |
| | % Cell viability [b] | | | | |
| F1: crude EtOH | ND | ND | 97.6±1.2 | 85.9±3.6 | 64.1±4.3 |
| F2: EtOH/Hex | ND | ND | 112.6±14.5 | 111.5±5.5 | 109.6±3.2 |
| F3: EtOH/EtOAc | ND | ND | 119.4±14.9 | 107.4±7.0 | 116.9±3.4 |
| F4: EtOH/BuOH | ND | ND | 90.2±10.0 | 46.6±5.3** | 13.3±0.6** |
| F5: EtOH/H₂O | ND | ND | 102.4±3.2 | 103.6±7.7 | 98.4±1.2 |
| ketotifen fumarate | 100.1±3.7 | 77.4±4.8 | ND | ND | ND |

The values are presented as means ± SEM (n = 3–6).

* $p < 0.05$,

** $p < 0.001$, compared with the vehicle control.

[a] 2- hour incubation time.

[b] 48-hour incubation time.

N/A indicates over 20% cytotoxicity was observed at 2- hour incubation time. ND means Not Determined.

**Table 2. Total phenolic and total flavonoid contents of the extracts of *X. tridentata*.**

| Sample | Total phenolic content | Total flavonoid content |
|---|---|---|
| | mg GAE/g extract | mg QE/g extract |
| F1: crude EtOH | 29.2±2.9** | 52.7±3.8** |
| F2: EtOH/Hex | 25.6±7.0** | 95.2±0.3** |
| F3: EtOH/EtOAC | 405.1±7.6 | 159.7±1.7 |
| F4: EtOH/BuOH | 136.9±6.5** | -3.2±0.6** |
| F5: EtOH/H₂O | 31.7±5.9** | -16.7±0.5** |

The values are presented as means ± SEM (n = 3).

$^{**}p < 0.001$, compared with F3: EtOH/EtOAc.

**Scheme 1. Preparation of the crude extract and sub-partitions from *X. tridentata*.**

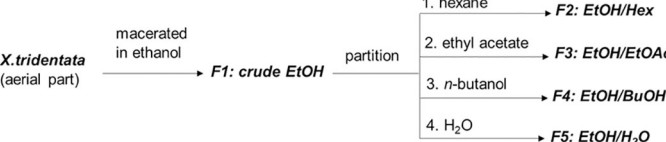

## Total phenolic and total flavonoid contents of *X. tridentata* extracts

According to previous reports on the high content of flavonoids and phenolic compounds present in *X. tridentata* and known anti-allergic activity of these classes of metabolites, preliminarily phytochemical screening on total phenolic and total flavonoid contents of the extracts was determined by Folin-Ciocalteu and aluminium chloride colorimetric methods, respectively. The results shown in Table 2 indicated that the EtOH/EtOAc fraction (F3) contained the highest amount of total phenolic and total flavonoid compounds, corresponding to its inhibitory potency on mast cell degranulation.

## Metabolomics analysis approach in identification of anti-allergic compounds present in the ethanol extracts of *X. tridentata*

LC-MS/MS metabolomics analysis was performed to identify active compounds responsible for anti-allergic activity. Firstly, five extracts (F1-F5) were analyzed with LC-MS/MS in both ESI positive and negative modes to obtain metabolite profiles (S1 Appendix). For data analysis, the MS raw files were subjected to peak alignment, peak picking, adduct grouping and normalization using Compound Discoverer 3.1 software (CD3.1). The parameters in CD3.1 for data analysis were set to be compatible with the chromatographic data obtained from this experiment. A retention time (RT) alignment step was obtained with a 5-ppm mass tolerance and a 2-min maximum shift. An unknown compound detection step was achieved with a 5-ppm mass tolerance, a signal to noise ratio of 3 and minimum peak intensity of 1x10⁴. Unknown compound grouping step was performed with a 5-ppm mass tolerance and a 0.2-min RT tolerance. The Area Under the Curve (AUC) of each metabolite was determined and normalized with that of pooled samples. Among 3,863 features obtained from the analysis, 2,860 features were detected only in positive ionization mode, 767 features were detected only in negative ionization mode, and 236 features were detected in both ionization modes. The data suggested that both positive and negative ionizations were needed for full metabolome coverage of the *X. tridentata* extracts.

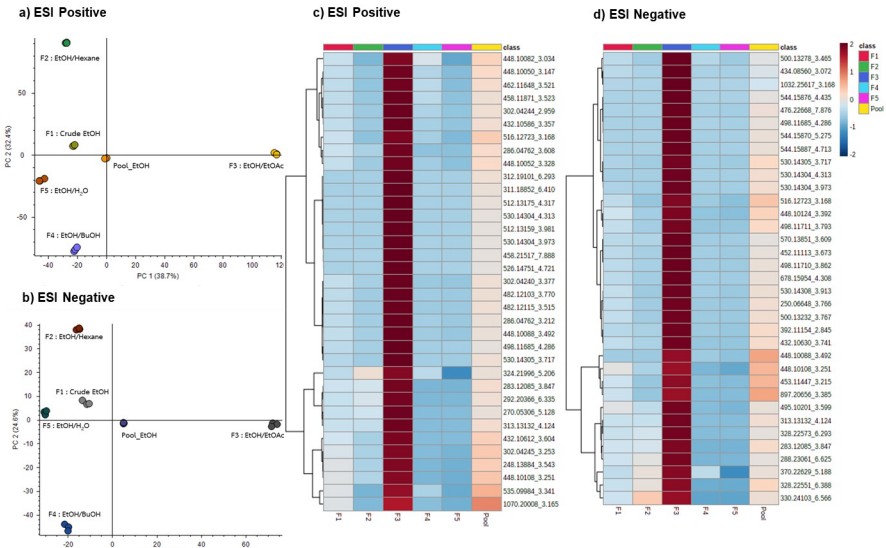

**Fig 2. Principle component analysis and heat map visualization of *X.tridentata* extracts metabolic profiles.**
Principal component 1 (PC1) represents the maximal variation of data. Principal component 2 (PC2) is orthogonal to
the PC1 axis and accounts for the second highest variation. a) Principle component analysis for ESI positive mode b)
Principle component analysis for ESI negative mode. Each point represents an individual LC-MS injection. c) Heat
map of 35 features with the highest fold change of F3/F5 observed in ESI positive mode and d) in ESI negative mode.

Principal Component Analysis (PCA) is a pattern recognition method used to visualize
general clustering trends. In PCA, data are not given or labeled with a primary classification
but clustered to find all unknown patterns. PCA is therefore an unsupervised analysis. In this
work, PCA was used to compare the chemical profiles of F1-F5 without classifying or pre-
grouping the samples. The score plots of each sample in positive and negative ionization
modes are shown in Fig 2a and 2b. Principal component 1 (PC1) represents the maximal varia-
tion of data, whereas principal component 2 (PC2) is orthogonal to the PC1 axis and accounts
for the second highest variation. PC1 and PC2 are the first and second principal components
accounted for 38.7% and 32.4% of the total variance in the data respectively in ESI positive
mode and explained for 48.3% and 24.6% of the total variance in the data respectively in ESI
negative mode. Herein, two-component PCA models accounted for the total variance of 71.1%
and 72.9% for ESI positive and negative, respectively. Each point represents an individual
LC-MS injection. The scatter of data dictates the similarities or differences of metabolites in
each sample. Samples containing similar metabolite content are congregated, while those con-
taining different metabolite content are disseminated in PCA. The result showed that all
pooled samples were closely clustered together demonstrating the reproducibility and robust-
ness of the LC-MS system. The metabolite profiles of the EtOH/EtOAc fraction (F3) and other
fractions (F1, F2, F4, and F5) were strongly discriminated in PCA in both positive and negative
ionization modes. The metabolites abundantly present in the F3 fraction were of our interest
since they potentially contributed to the strong discrimination in PCA and were possibly
attributed to the anti-allergic activity.

To determine the features with significant association to the anti-allergic activity, the meta-
bolomics data analysis focused on features that were high abundant in the F3 fraction and low
abundant in the F5 fraction, since F3 and F5 possessed the highest and lowest anti-allergic
activities, respectively. In the first filtering step using % coefficient of variation (%CV) less

than 30%, 3,863 normalized ions were determined. The magnitude of difference between the two sample groups, so called fold change (FC), was then applied in the second filtering step. The FC of AUC (Area Under the Curve) of F3 *vs* F5 ($AUC_{EtOH/EtOAc}$/ $AUC_{EtOH/H2O}$) was used to determine the significant features. As a result, top 35 features (FC = 590 for data in ESI positive mode and FC = 280 for data in ESI negative mode) were determined and then selected for hierarchical clustering to discover trends within the partition samples (Fig 2c and 2d). In both positive and negative ionization modes, all features were predominantly presented in the F3 fraction and minimally observed in other fractions.

To facilitate the compound annotation step, an in-house mass list of reported compounds found in *Xenostegia tridentata* (L.) D.F. Austin & Staples or plant species in the Convolvulaceae family (*Merremia tridentata*, *Ipomoea carnea*, *Evolvulus alsinoides*, *Evolvulus nummularius*, *Ipomoea batatas*, *Argyreia speciose* and *Argyreia cuneate*) was constructed. The literature curation was performed by searching the following keywords: "*Xenostegia tridentata*", "*Merremia tridentata*", "*Convolvulaceae*", and "*Thao tot ma*". The list contains 116 compounds (S2 Table) with information including compound name, chemical formula, molecular weight, m/z, chemical structure, InChIKey, CAS number, IUPAC name, common name, Chem Spider ID, ChemSpider link, reported plant species with reference publications and reported activities with references. Reported mass spectroscopic information on public databases (mzCloud and METLIN) was also included to facilitate further compound annotation with MS techniques.

In this work, the LC-MS data acquisition was designed to obtain both MS1 and MS2 data simultaneously. MS1 data from high-resolution mass spectrometer (< 5 ppm mass accuracy) were used to predict the chemical formula and match it against MS1 public databases, including ChemSpider, NIST Chemistry WebBook, Phenol-Explorer, Planta Piloto de Química Fina, Universidad de, PlantCyc and Sigma-Aldrich. In addition, various MS2 fragmentations were performed to increase the confidence of the annotation step. The MS2 spectra were matched with the mzCloud database to aid compound annotation (S3 Table). The significant features that contributed to the discrimination between F3 and F5 fractions were identified according to the metabolomics standard initiative (MSI) [43]. MSI level 1–4 were proposed as a minimal reporting standard for chemical analysis. MSI level 4 refers to unidentified or unclassified compounds that can be detected using spectral data. MSI level 3 includes compounds that can be putatively characterized or classified by using physicochemical properties or spectral similarity to a chemical class. MSI level 2 is for putatively annotated compounds with spectral similarity to known compounds in public/commercial spectral libraries. MSI level 1 is for novel and non-novel identified compounds. Novel MSI I metabolites can be characterized by elemental analysis, NMR, IR, UV and accurate mass measurement, while non-novel MSI I metabolite identification can be validated with commercial standard compounds by comparison of a minimum of two independent and orthogonal data under identical experimental conditions.

LC-MS metabolite profiles of five extracts (F1-F5) in both ESI positive and negative (S1 Appendix) were subjected to metabolomics data analysis as previously described. The LC-MS profile is a base peak ion chromatogram showing the relative abundance of peaks presented in each sample. The major peak for five extracts analyzed in ESI positive mode at 5.15–5.22 min in S1 Appendix was predicted to be a compound with a molecular weight of 324.21996 and a chemical formula of $C_{21}H_{28}N_2O$. For compound identification, MS2 showed no match to any spectrum on mzCloud database. Based on MS1 data, this major peak at 5.15–5.22 min could be predicted with the lowest mass error of -0.64 ppm to several compounds including 4,4'-bis (diethylamino)benzophenone, 1-(9H-carbazol-9-yl)-3-(dipropylamino)-2-propanol, N-[4-(diethylamino)phenyl]-4-(2-methyl-2-propanyl)benzamide, diampromide and N2-butyl-N-(1,2-diphenylethyl)-N2-methylglycin-amide. The significant features in positive and negative ionization modes that contributed to the discrimination between F3 and F5 fractions were

**Table 3. Summary of metabolite features contributed to the discrimination between F3: EtOH/EtOAc and F5: EtOH/H₂O fractions in ESI positive mode.**

| NO. | ESI | FEATURE | PREDICTED FORMULA | PUTATIVE IDENTIFIED COMPOUND[1] |
|-----|-----|---------|-------------------|-------------------------------|
| 1 | Pos | 248.13884_3.543 | C11 H16 N6 O | |
| 2 | Pos | 270.05306_5.128 | C15 H10 O5 | Apigenin[2] |
| 3 | Pos | 283.12085_3.847 | C17 H17 N O3 | (2E)-3-(4-Hydroxyphenyl)-N-[2-(4-hydroxyphenyl)ethyl]acrylamide[2] |
| 4 | Pos | 286.04762_3.608 | C15 H10 O6 | Flavonoids[3] |
| 5 | Pos | 286.04762_3.212 | C15 H10 O6 | Flavonoids[3] |
| 6 | Pos | 292.20366_6.335 | C18 H28 O3 | |
| 7 | Pos | 302.0424_3.377 | C15 H10 O7 | Flavonoids[3] |
| 8 | Pos | 302.04244_2.959 | C15 H10 O7 | Flavonoids[3] |
| 9 | Pos | 302.04245_3.253 | C15 H10 O7 | Flavonoids[3] |
| 10 | Pos | 311.18852_6.41 | C14 H26 N5 O P | |
| 11 | Pos | 312.19101_6.293 | C13 H24 N6 O3 | |
| 12 | Pos | 313.13132_4.124 | C18 H19 N O4 | |
| 13 | Pos | 324.21996_5.206 | C21 H28 N2 O | |
| 14 | Pos | 432.10586_3.357 | C21 H20 O10 | Apigetrin (Apigenin-7-O-glucoside)[2] |
| 15 | Pos | 432.10612_3.604 | C21 H20 O10 | Afzelin (Kaempferol-3-O-rhamnoside)[2] |
| 16 | Pos | 448.1005_3.147 | C21 H20 O11 | Flavonoid glycosides[3] |
| 17 | Pos | 448.10052_3.328 | C21 H20 O11 | Flavonoid glycosides[3] |
| 18 | Pos | 448.10082_3.034 | C21 H20 O11 | Cynaroside (luteolin-7-O-glucoside)[2] |
| 19 | Pos | 448.10088_3.492 | C21 H20 O11 | Flavonoid glycosides[3] |
| 20 | Pos | 448.10108_3.251 | C21 H20 O11 | Quercitrin (Quercetin-3-O-rhamnoside)[2] |
| 21 | Pos | 458.11871_3.523 | C10 H26 N4 O14 S | |
| 22 | Pos | 458.21517_7.888 | C23 H30 N4 O6 | |
| 23 | Pos | 462.11648_3.521 | C22 H22 O11 | Methoxy flavonoid glycosides[3] |
| 24 | Pos | 482.12103_3.77 | C25 H22 O10 | Silymarin (flavonolignans)[3] |
| 25 | Pos | 482.12115_3.515 | C25 H22 O10 | Silymarin (flavonolignans)[3] |
| 26 | Pos | 498.11685_4.286 | C26 H18 N4 O7 | |
| 27 | Pos | 512.13159_3.981 | C22 H20 N6 O9 | |
| 28 | Pos | 512.13175_4.317 | C23 H16 N10 O5 | |
| 29 | Pos | 516.12723_3.168 | C25 H24 O12 | Dicaffeoylquinic acid[3] |
| 30 | Pos | 526.14751_4.721 | C27 H26 O11 | Viscutin 1[3] |
| 31 | Pos | 530.14304_4.313 | C26 H26 O12 | Flavonoid glycoside derivatives[3] |
| 32 | Pos | 530.14304_3.973 | C26 H26 O12 | Flavonoid glycoside derivatives[3] |
| 33 | Pos | 530.14305_3.717 | C26 H26 O12 | Flavonoid glycoside derivatives[3] |
| 34 | Pos | 535.09984_3.341 | C16 H23 N7 O10 P2 | |
| 35 | Pos | 1070.20008_3.165 | C43 H53 N4 O18 P3 S2 | |

[1] Putative identified compounds were determined as level 1–4 according to the Metabolomics Standard Initiative (MSI).

[2] Compounds with MSI level 2 were putatively annotated based on spectral similarity to available databases.

[3] Compounds with MSI level 3 that putatively characterized by their compound classes according to spectral similarity to known compounds in that chemical class.

[4] Unknown compounds with MSI level 4 can be distinguished from spectra data but remain unclassified or unidentified based on MS and MS/MS data.

summarized in Tables 3 and 4, respectively. The identification level of significant features was determined by following MSI guidelines. For example, feature 270.05306_5.128 showed a high percentage of MS2 matching to several compounds on mzCloud (S3 Table). This feature showed the highest matching score to apigenin in both positive and negative ionization modes. It is noted that the observed m/z 117.0346 at HCD 40 under ESI negative mode was only matched with similar intensity and pattern to that of apigenin but not to other

**Table 4. Summary of metabolite features contributed to the discrimination between F3: EtOH/EtOAc and F5: EtOH/H$_2$O fractions in ESI negative mode.**

| NO. | ESI | FEATURE | PREDICTED FORMULA | PUTATIVE IDENTIFIED COMPOUND |
|---|---|---|---|---|
| 1 | Neg | 250.06648_3.766 | C13 H14 O3 S | |
| 2 | Neg | 283.12085_3.847 | C17 H17 N O3 | (2E)-3-(4-Hydroxyphenyl)-N-[2-(4-hydroxyphenyl)ethyl]acrylamide[2] |
| 3 | Neg | 288.23061_6.625 | C16 H32 O4 | |
| 4 | Neg | 313.13132_4.124 | C18 H19 N O4 | |
| 5 | Neg | 328.22551_6.388 | C18 H32 O5 | Corchorifatty acid F[2] |
| 6 | Neg | 328.22573_6.293 | C18 H32 O5 | Corchorifatty acid F[2] |
| 7 | Neg | 330.24103_6.566 | C18 H34 O5 | |
| 8 | Neg | 370.22629_5.188 | C20 H35 O4 P | |
| 9 | Neg | 392.11154_2.845 | C19 H20 O9 | |
| 10 | Neg | 432.1063_3.741 | C21 H20 O10 | Flavonoid glycosides[3] |
| 11 | Neg | 434.0856_3.072 | C20 H18 O11 | Flavonoid glycosides[3] |
| 12 | Neg | 448.10088_3.492 | C21 H20 O11 | Flavonoid glycosides[3] |
| 13 | Neg | 448.10108_3.251 | C21 H20 O11 | Quercitrin (Quercetin-3-O-rhamnoside)[2] |
| 14 | Neg | 448.10124_3.392 | C21 H20 O11 | Flavonoid glycosides[3] |
| 15 | Neg | 452.11113_3.673 | C24 H20 O9 | Flavan-3-ol derivatives[3] |
| 16 | Neg | 453.11447_3.215 | C16 H20 N7 O7 P | |
| 17 | Neg | 476.22668_7.876 | C23 H32 N4 O7 | |
| 18 | Neg | 495.10201_3.599 | C22 H17 N5 O9 | |
| 19 | Neg | 498.11685_4.286 | C26 H18 N4 O7 | |
| 20 | Neg | 498.1171_3.862 | C26 H18 N4 O7 | |
| 21 | Neg | 498.11711_3.793 | C22 H14 N10 O5 | |
| 22 | Neg | 500.13232_3.767 | C25 H24 O11 | |
| 23 | Neg | 500.13278_3.465 | C25 H24 O11 | |
| 24 | Neg | 516.12723_3.168 | C25 H24 O12 | Dicaffeoylquinic acid[3] |
| 25 | Neg | 530.14304_4.313 | C26 H26 O12 | Flavonoid glycoside derivatives[3] |
| 26 | Neg | 530.14304_3.973 | C26 H26 O12 | Flavonoid glycoside derivatives[3] |
| 27 | Neg | 530.14305_3.717 | C26 H26 O12 | Flavonoid glycoside derivatives[3] |
| 28 | Neg | 530.14308_3.913 | C26 H26 O12 | Flavonoid glycoside derivatives[3] |
| 29 | Neg | 544.1587_5.275 | C28 H24 N4 O8 | |
| 30 | Neg | 544.15876_4.435 | C29 H20 N8 O4 | |
| 31 | Neg | 544.15887_4.713 | C28 H24 N4 O8 | |
| 32 | Neg | 570.13851_3.609 | C23 H23 N8 O8 P | |
| 33 | Neg | 678.15954_4.308 | C34 H30 O15 | 3,4,5-tri-O-caffeoylquinic acid[3] |
| 34. | Neg | 897.20656_3.385 | C37 H41 N9 O12 P2 S | |
| 35 | Neg | 1032.25617_3.168 | C52 H40 N8 O16 | |

[1] Putative identified compounds were determined as level 1–4 according to the Metabolomics Standard Initiative (MSI).

[2] Compounds with MSI level 2 that were putatively annotated based on spectral similarity to available databases.

[3] Compounds with MSI level 3 that were putatively characterized by their compound classes according to spectral similarity to known compounds in that chemical class.

[4] Unknown compounds with MSI level 4 that can be distinguished from spectra data but remain unclassified or unidentified based on MS and MS/MS data.

compounds including genistein. Using metabolomics data, feature 270.05306_5.128 was therefore putatively identified as apigenin with MSI level 2. To increase the level of identification to MSI level 1, commercial apigenin will be needed for comparing RT, MS1 and MS2. In another example, with MS1 information, feature 516.12723_3.168 could be 4,5-dicaffeoylquinic acid, 3,5-dicaffeoylquinic acid, 3,4-di-O-caffeoylquinic acid, or 1,3-di-O-caffeoylquinic acid. However, only one hit of 4,5-dicaffeoylquinic acid was obtained from MS2 matching (S3 Table).

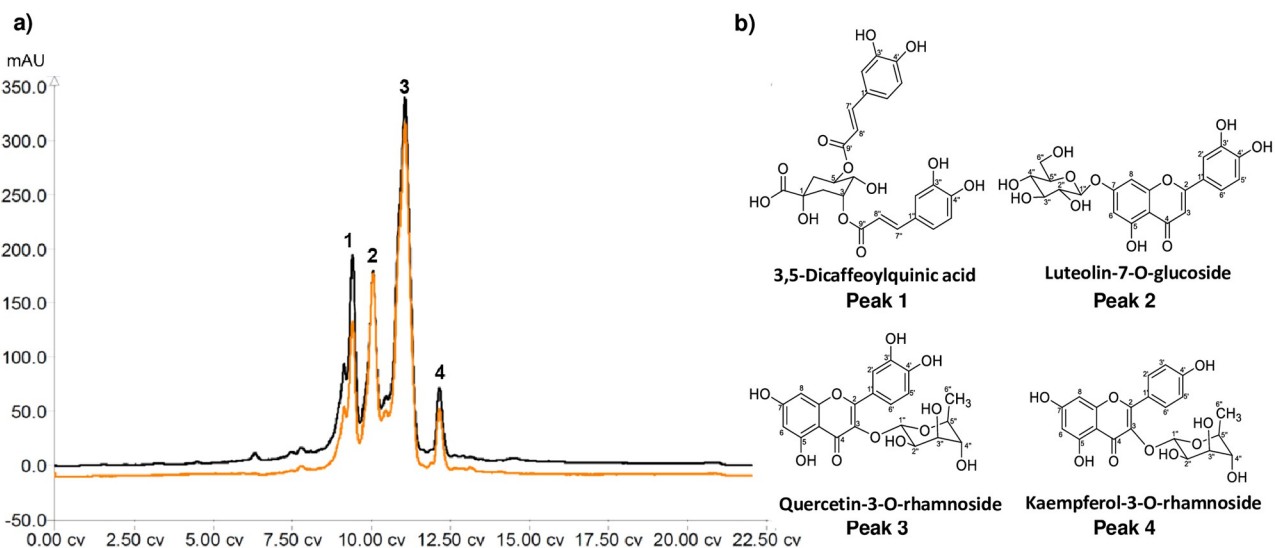

**Fig 3. Isolation of major compounds from F3: EtOH/EtOAc.** a) Chromatogram of subfraction F3 from MPLC separation on a reversed phase silica gel column eluted with gradient system of MeOH and H$_2$O (3–97%). The chromatographic signals at 256 nm (black) and 366 nm (orange) were detected. b) Chemical structures of the known compounds corresponding to peak1-4 (see S1 and S2 Appendices for the HRMS and NMR spectra, respectively).

This result was potentially from the fact that 4,5-dicaffeoylquinic acid was the only isomer identified from both positive and negative MS2 spectra on mzCloud database. Therefore, with metabolomics data, feature 516.12723_3.168 was putatively identified as dicaffeoylquinic acid with MSI level 3. To increase the level of identification shown in Tables 3 and 4 to MSI level 1, either standard or purified compounds would need to be analyzed to ensure matching RT, MS1 and MS2 under good separation conditions. As a result, fraction F3 was further purified and the resulting pure compounds were subjected to NMR experiments to confirm the chemical structures.

## Validation of chemical structures of the significant metabolites

Among the sub-partitions, F3 contained the highest contents of flavonoid and phenolic compounds and exhibited the highest inhibitory activity on mast cell degranulation. Further purification of this fraction was therefore performed. One gram of F3 was subjected to the MPLC separation. The chromatogram of F3 displays four major peaks (Fig 3). The collected pure compounds of each peak were subjected to LC-MS/MS under the same conditions as the metabolite profiling experiment. The accurate mass, MS2 fragmentation patterns with various energies and RT were well-matched to mass spectra of the F3: EtOH/EtOAc (S1 Appendix). The results showed that peak 1–4 were features 516.12723_3.168, 448.10082_3.034, 448.10108_3.251 and 432.10612_3.604, respectively. The four features obtained from metabolite profiling and the isolated compounds were in good agreement with only 0.02 min of RT difference and 3 ppm of mass difference. The NMR and HRMS analysis of the collected fractions of each peak revealed that peak 1–4 corresponded to four known compounds, 3,5-dicaffeoylquinic acid (0.06 g, 6.3%), luteolin-7-O-glucoside (0.09 g, 8.8%), quercetin-3-O-rhamnoside (0.15 g, 14.8%) and kaempferol-3-O-rhamnoside (0.03 g, 3.1%), respectively. Relative metabolite levels were obtained from the average area under the curve (AUC) of each feature in the extracts F1-F5, as shown in Fig 4. The level of the four identified metabolites was

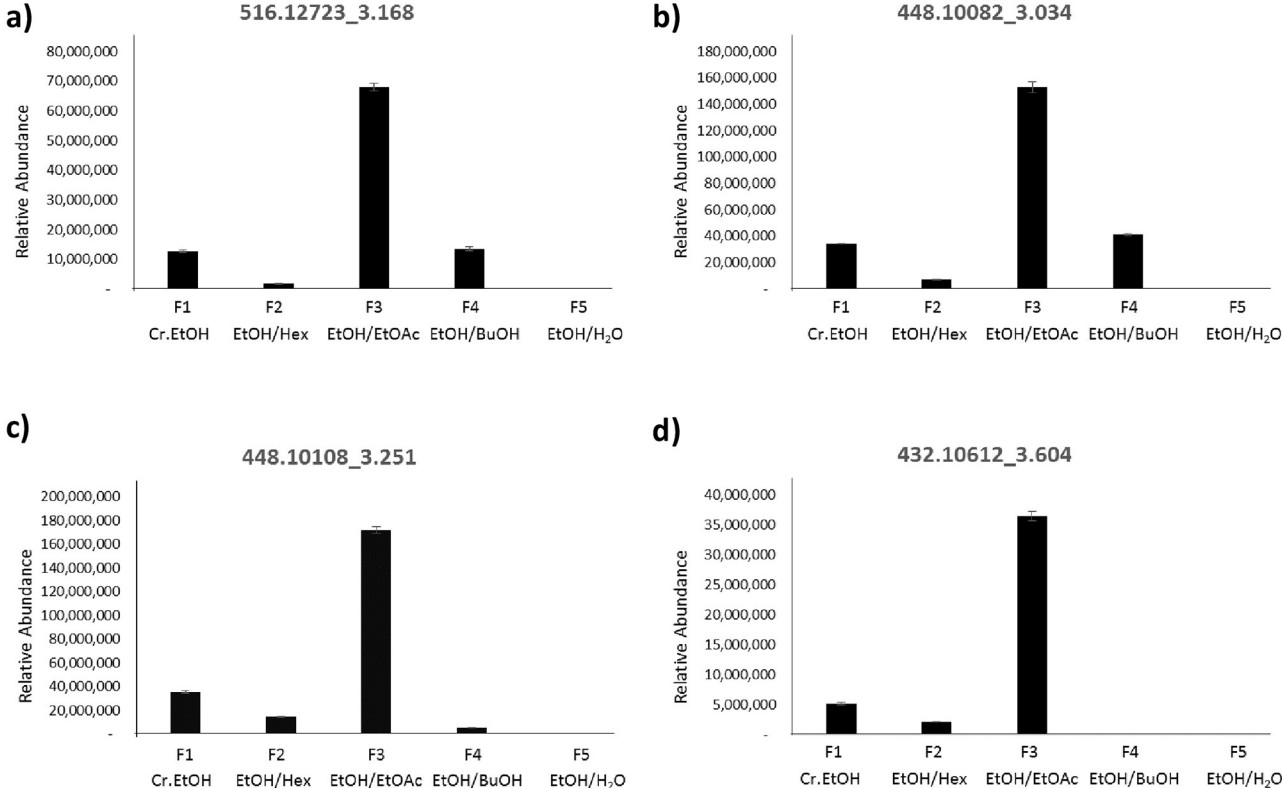

**Fig 4. The levels of four significant metabolite features in different subfractions.** a) ESI+516.12723_3.168, b) ESI+448.10082_3.034, c) ESI
+448.10108_3.251 and d) ESI+432.10612_3.604.

significantly elevated in the F3 fraction potentially suggesting their contribution to the anti-allergic activity.

## Validation of anti-allergic activity of the isolated pure compounds

The phenolic and flavonoid compounds identified from the metabolomics analysis have been widely reported for their bioactivities including anti-allergic activities [10, 44–51]. For example, 3,5-dicaffeoylquinic acid, quercetin-3-O-rhamnoside and luteolin-7-O-glucoside exhibited dose-dependent *in vitro* and *in vivo* inhibitions of mast cell degranulation induced by compound 48/80, a non-IgE-mediated mast cell degranulation agent [44, 50, 51]. Kaempferol-3-O-rhamnoside showed inhibitory effects on OVA-induced airway allergic inflammation *in vivo* [52]. To validate the anti-allergic activities of these compounds in our assay, the IgE-mediated and compound 48/80-stimulated mast cell degranulation assays were performed. The results shown in Figs 5b and 6a confirmed the inhibitory activities of the four isolated compounds and ketotifen fumarate on the degranulation activated by both stimulants.

Generation of intracellular reactive oxygen species (ROS) within mast cells upon activation with chemical agents or antigens has been known to play important roles in the production of proinflammatory cytokines and degranulation [53, 54]. Attenuation of ROS generation could therefore impair degranulation and subsequently diminish the allergic cascade. To examine the inhibitory effect of *X. tridentata* extracts on ROS production, RBL-2H3 cells were loaded with DCFH-DA, a redox-sensitive dye, incubated with *X. tridentata* extracts and activated

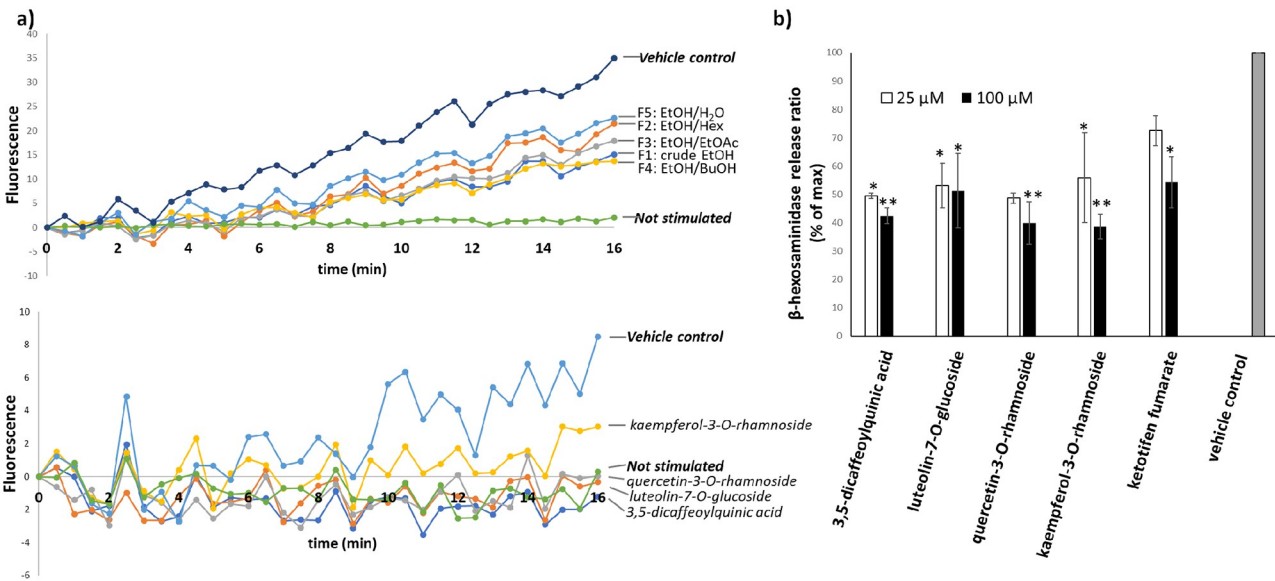

**Fig 5. Inhibitory effects on IgE-mediated RBL-2H3 cell degranulation.** a) The effects of the extracts (100 μg/mL) and active compounds (25 μM) derived from *X. tridentata* on intracellular ROS generation b) The effects of active compounds and ketotifen fumarate (25 and 100 μM) on β-hexosaminidase release. Ketotifen fumarate was used as a positive control. The values are presented as means ± SEM (n = 3). $^{*}p<0.05$ and $^{**}p < 0.001$, compared with the vehicle control.

with anti-DNP-IgE and DNP-BSA or compound 48/80. After activation, the fluorescence signal, representative of ROS generation, was measured for 16 minutes using a microplate reader (30-second intervals). The results shown in Figs 5a and 6b indicate that the extracts and the four isolated compounds could suppress ROS generation in the RBL-2H3 cells activated by IgE-dependent activation or compound 48/80.

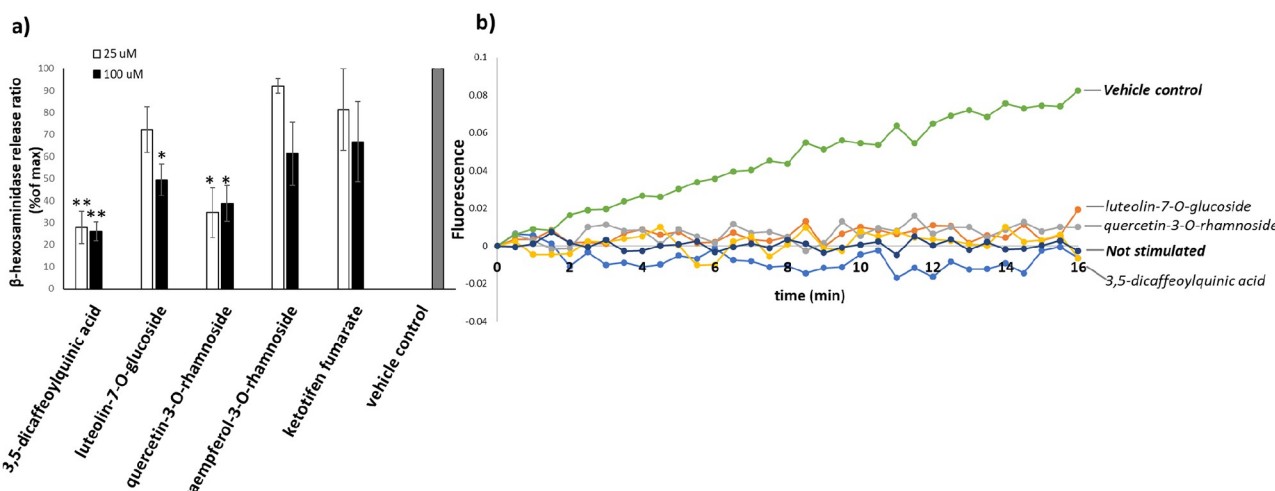

**Fig 6. Inhibitory effects on compound 48/80-stimulated RBL-2H3 cell degranulation.** a) The effects of the active compounds and ketotifen fumarate (25 and 100 μM) on β-hexosaminidase release. Ketotifen fumarate was used as a positive control. The values are present as means ± SEM (n = 4). $^{*}p<0.05$ and $^{**}p < 0.001$, compared with the vehicle control. b) The effects of active compounds (25 μM) derived from *X. tridentata* on intracellular ROS generation.

In summary, crude ethanol extract *of X. tridentata* exhibited *in vivo* and *in vitro* inhibitory effects on IgE-mediated mast cell degranulation, a process relating to type I hypersensitivity. Among the subfractions, the ethyl acetate subfraction (F3) showed the highest inhibitory activity on IgE-mediated RBL-2H3 cell degranulation, presumably due to its high flavonoid and phenolic contents. From LC-MS/MS metabolomics analysis, luteolin-7-O-glucoside, quercetin-3-O-rhamnoside, kaempferol-3-O-rhamnoside and 3,5-dicaffeoylquinic acid were listed among the top 35 features that were found in more abundance in F3 than in other subfractions. These four compounds corresponded to the four major metabolites isolated from F3. The chemical structures were confirmed by NMR and HRMS techniques. As reported previously, the isolated compounds showed inhibitory effects on mast cell degranulation and intracellular ROS generation, assuring their important roles in the anti-allergic property of *X tridentata*. Notably, this work highlights the utilization of LC-MS/MS metabolomics analysis as an efficient tool for active metabolite identification in plants. Moreover, anti-allergic activity of *X.tridentata* and the presence of quercetin-3-O-rhamnoside, kaempferol-3-O-rhamnoside and 3,5-dicaffeoylquinic in this plant are reported. To apply this plant as complementary or alternative medicine for allergic diseases, studies on molecular responses according to the extracts and the active components are underway in our laboratory to fulfill understanding of their modes of action.

## Supporting information

**S1 Appendix. LC-MS metabolite profiling and comparison of MS and MS/MS data of the features obtained from metabolite profiling to isolated compounds.**
(PDF)

**S2 Appendix. NMR spectra of the isolated compounds.**
(PDF)

**S1 Table. Extraction yields, β-hexosaminidase inhibitory activity of the extracts, cytotoxicity of the isolated compounds and previously reported anti-allergic activities of the isolated compounds.**
(PDF)

**S2 Table. Library for LC-MS.**
(XLSX)

**S3 Table. Metabolite identification of features contributed to the discrimination between F3:EtOH/EtOAc and F5:EtOH/H$_2$O fractions.**
(PDF)

## Acknowledgments

The authors would like to extend sincerest appreciation to Dr. Pranee Nangam at Faculty of Science, Naresuan University, Thailand, for her kind help on plant species identification and Program in Environmental Toxicology, Chulabhorn Graduate Institute, for providing the equipment for the cell culture experiments. The authors gratefully acknowledge Dr. Ubolsree Lertsakulpanich, Dr. Suganya Yongkiettrakul, and Dr. Warangkhana Songsungthong at National Center for Genetic Engineering and Biotechnology (BIOTEC), National Science and Technology Development Agency (NSTDA) and Mrs. Suwalya Khemvaraporn at Thammasat University for guidance and manuscript revision.

## Author Contributions

**Conceptualization:** Jaruwan Chatwichien.

**Formal analysis:** Rinrada Suntivich, Worawat Songjang, Arunya Jiraviriyakul, Jaruwan Chatwichien.

**Funding acquisition:** Jaruwan Chatwichien.

**Investigation:** Rinrada Suntivich, Worawat Songjang, Arunya Jiraviriyakul, Jaruwan Chatwichien.

**Methodology:** Rinrada Suntivich, Worawat Songjang, Arunya Jiraviriyakul, Jaruwan Chatwichien.

**Project administration:** Jaruwan Chatwichien.

**Resources:** Jaruwan Chatwichien.

**Supervision:** Somsak Ruchirawat.

**Validation:** Jaruwan Chatwichien.

**Writing – original draft:** Rinrada Suntivich, Worawat Songjang, Arunya Jiraviriyakul, Jaruwan Chatwichien.

**Writing – review & editing:** Rinrada Suntivich, Worawat Songjang, Arunya Jiraviriyakul, Somsak Ruchirawat, Jaruwan Chatwichien.

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
