## [Decision Letter · Decision Letter 0]

26 Oct 2021

PONE-D-21-29347LC-MS/MS metabolomics-facilitated identification of the active compounds responsible for anti-allergic activity of the ethanol extract of Xenostegia tridentataPLOS ONE

Dear Dr. Chatwichien,

Thank you for submitting your manuscript to PLOS ONE. After careful consideration, we feel that it has merit but does not fully meet PLOS ONE’s publication criteria as it currently stands. Therefore, we invite you to submit a revised version of the manuscript that addresses the points raised during the review process.

Please be careful when  providing information about the studied species, since several synonyms have been and are still in use. Moreover, do introduce these synonyms in the Introduction section. Some of methodological approaches used in the study are not properly performed as being pointed out in the Reviewer's report. These concerns should be meticulously checked and additional experiments performed where needed. Statistical analyzes also deserve more authors' attention. Language usage should be considerably improved. It is highly recommended to have the manuscript read and copy-edited by a native English speaker or a professional editing agency.

We look forward to receiving your revised manuscript.

Kind regards,

Branislav T. Šiler, Ph.D.

Academic Editor

PLOS ONE

Journal Requirements:

Reviewers' comments:

Reviewer's Responses to Questions

**Comments to the Author**

1. Is the manuscript technically sound, and do the data support the conclusions?

Reviewer #1: Partly

2. Has the statistical analysis been performed appropriately and rigorously? 

Reviewer #1: Yes

3. Have the authors made all data underlying the findings in their manuscript fully available?

Reviewer #1: Yes

4. Is the manuscript presented in an intelligible fashion and written in standard English?

Reviewer #1: Yes

5. Review Comments to the Author

Reviewer #1: Authors describe the anti-allergic effects of plant Merrenia (X.) tridentata in vitro and in vivo and use LC-MS/MS to identify several compounds with the following isolation to confirm the identity and the activity, as well in relation to the references. The work is interesting and well designed, with the logical sequence. The metabolomic approach is interesting, but the lack of proper library data makes the actual compound identification challenging. It is valuable that authors isolated the major compounds and tested them for activity as well. The data however, often show limitations and some parts are not clear. Data statisticts and viability should be concerned, following also other comments below:

1. “Xenostegia tridentata (L.) D.F. Austin & Staples is a synonym of Merremia tridentata(L.) Hallier f.”, please see below link and correct the statements about the first isolations from the plant (e.g., the last paragraph “for the first time” is not appropriate, because many previous studies of Merrenia were shown, also in the end of Abstract P. 2. Line 15). P. 3 please mention the synonym of the plant. For plant synonyms, please check theplantlist.org. here: http://www.theplantlist.org/tpl1.1/record/tro-8502902

2. Timing of the assays. The errors of the assays, SEM show high error. The viability window for F4 (BuOH) and ketotifen doesn’t fit the antigen-induced assay. Ketotifen is approx. 25% toxic at 100uM which would seriously affect the assay response considering sensitivity of antigen-induced assay in RBL cells (unless the timing of treatments was significantly different which is not shown in the Table 1)? Was the samples treatment time in Table 1. For MTT 48 hr and for beta-hexo assay 2hr? Please describe and explain why the cytotoxic samples (including kerotifen at 100uM with over 20% toxicity) were used in antigen-induced beta-hexo assay and the effects of butanol layer are not fitting their toxic levels? Is it because of the different time treatment?

3.

4. In the procedure, long incubation times may cause statistical errors particularly in the antigen induced assays.

5. P. 6. Line 10 Was the solubility of extracts in tween saline solution good enough? Was the sample filtered before administration? Was the solvent and vehicle the same? Please describe clearly.

6. Isolation scheme and Table 1. Considering that all the samples were oily, and the major fraction was hexane layer (11 g), while ethyl acetate layer was the minor (0.9 g) one. Referring also to Table 4 that content of these compounds is low in crude extract, while 3.034 is high in BuOH. => Is it possible that the most active fraction is hexane layer and maybe due to its unsaturated fatty acid content which exerts strong anti-inflammatory properties? It would be useful to investigate more on the content of this fraction.

7. Why Compound 48/80 was chosen as inducer, what is its mechanism of action?

8. P. 21. Line 13 Mention the level of activity, IC50 or values would be useful as well as the comparison with literature values. There was no concentration manner observed from Fig. 5 and 6 for pure compounds activity, is it reasonable I comparison with literature??

9. Vehicle control in Fig. 4 ROS is very unstable.

10. Ketotifen at 100 (ug/mL or uM?) should not be used if was toxic by over 20% in Table 1. And why it’s values differ so much from Fig. 5 and 6 (35 vs 60 for 100ug/ml or uM?). The concentration unit is also unclear?

11. Why MS2 in mzcloud give sometimes no hit, while often many hits, which fragment (product) ion was considered?

12. Please explain abbreviations: What does NSS, ICR mean.

13. Figure 1 The description of in vivo results may be clarified. The fact that the high concentration of extract was not inhibiting PCA in the same manner as control dexamethasone (also because the concentration used was different). Even the top concentration of extract did not statistically inhibit PCA. Is it due to a statistical error or is similar concentration effect observed in vitro as well? The cause may be too short pre-treatment in vivo with only a single dose??

14. Table 1. The statistics is not reasonable. Stars are missing for ketotifen at 100 and crude or water at 2000 ug/mL. It is not reasonable to evaluate the effects of the samples toxic to the cells, such as when the cell death is over 20%, especially for butanol layer. Also the concentrations used were extremely high. Is there a reasoning for it? It would be useful to see thte IC50 values for clarification in the text, because the current description is very brief and often skips the inconsistent results.

15. Is the beta-hexo release ratio percentage of max, meaning max is triton or inducer alone?

16. What is AGC and it’s value of 1x105?

17. How authors could use 1 g of EA fraction when only 0.9 g was obtained?

18. P. 13. Line 18 The statement “The parameters were adjusted to fit to the chromatographic data obtained from this experiment” it’s not clear, how the parameters were adjusted and if in objective manner?

19. Do the features represent the mass?

20. How the authors ensure that the isolated compounds were matching the metabolomics study?

21. Except the in-house mass list, was there another database available and used for MS2 except mzcloud which often gave no result?

22. Metabolomics Standard Initiative MSI levels 1-4 and the differences between them should be described. The resulting compounds identification such as four isolated compounds or apigenin should be given.

23. The effects of compounds on viability and hexosaminidase enzymatic activity should be mentioned clearly in the text (currently in Sup data), while negative values of EA layer should be explained, and methodology reviewed, -46.1 ± 0.9 (1000ug/mL) -89.5 ± 0.5 (2000 ug/mL)

24. P. 29. Line 8-9. The condition should be mentioned in Materials section. The further purification besides MPLC should be described and purity level estimated. The Fig. 3 MPLC chromatogram peaks are overlapping.

25. What represents the last four number of each “feature”?

26. Please unify a reference style, there are missed capital for Genus name or the style is inconsistent. Capitalization, please unify

27. LC-MS/MS in S1 show only few peaks, on which basis they were identified and what could be the major peak at 5.17?

There are many syntax and grammar errors or misspellings, please see following as examples only:

P. 6 l. 12 by intraperitoneal administration

What does NSS, ICR mean.

P13, l. 3 screening was

P.14, l. 9 discrimination possibly be attributed to

P.3 lines 21-25 rephrase please. Statistic treatment,

P. 4 knowledges.

P. 4 Line 13 contributed

P. 21. Line 6 bioassay-guided is not so clear

P. 13. Line 16 detect

P. 14 Unsupervised analysis could be explained. Figure 2 is not very clear.

P. 15. Line 1-2 rephrase please.

P. 29 cpd 24 name

Conclusion. Please specify statements: P. 23. Line 2 “Some proinflammatory factors” is too vague

Line 8-10 please rephrase

Line12-15 “to its highly abundance” rephrase and consider about role of hexane layer

Line14-16 please rephrase

PCA (same for in vivo experiment) abbreviation should not be used for “Fig 2. Principle component analysis (PCA) and Heat map visualization of X.tridentata extracts.”

Also, the analysis and discrimination of layers is not clear.

6. PLOS authors have the option to publish the peer review history of their article (what does this mean?). If published, this will include your full peer review and any attached files.

Reviewer #1: **Yes: **Michal Korinek

---

## [Author Response · Author response to Decision Letter 0]

9 Dec 2021

Please kindly find our responses to reviewers, detail about our revision and our explanation to the reviewer’s concerns in the “response to reviewers” file.

---

## [Decision Letter · Decision Letter 1]

10 Jan 2022

PONE-D-21-29347R1LC-MS/MS metabolomics-facilitated identification of the active compounds responsible for anti-allergic activity of the ethanol extract of Xenostegia tridentataPLOS ONE

Dear Dr. Chatwichien,

Thank you for submitting your manuscript to PLOS ONE. After careful consideration, we feel that it has merit but does not fully meet PLOS ONE’s publication criteria as it currently stands. Therefore, we invite you to submit a revised version of the manuscript that addresses the points raised during the review process.

Although the authors have amended the manuscript according to the Reviewer's report, these is still a room for improvement. Please meticulously check the new report and supplement the manuscript with the necessary information.

We look forward to receiving your revised manuscript.

Kind regards,

Branislav T. Šiler, Ph.D.

Academic Editor

PLOS ONE

Journal Requirements:

Reviewers' comments:

Reviewer's Responses to Questions

**Comments to the Author**

1. If the authors have adequately addressed your comments raised in a previous round of review and you feel that this manuscript is now acceptable for publication, you may indicate that here to bypass the “Comments to the Author” section, enter your conflict of interest statement in the “Confidential to Editor” section, and submit your "Accept" recommendation.

Reviewer #1: All comments have been addressed

2. Is the manuscript technically sound, and do the data support the conclusions?

Reviewer #1: Yes

3. Has the statistical analysis been performed appropriately and rigorously? 

Reviewer #1: Yes

4. Have the authors made all data underlying the findings in their manuscript fully available?

Reviewer #1: Yes

5. Is the manuscript presented in an intelligible fashion and written in standard English?

Reviewer #1: Yes

6. Review Comments to the Author

Reviewer #1: Authors addressed the comments in comprehensive manner and although there are limitations such as the high doses used in bioassays (the plant is not food), low ratio of identified components based on the metabolomics study, the overall quality was improved and the manuscript is suitable for publication. The approach authors chose may serve as a template for future discovery of antiallergic drugs from herbal medicines.

Minor comments:

1. The role of LC-MS/MS metabolomics in current study may be exaggerated since the methods failed to identify the major compounds in the first place.

2. The PC 1 and PC 2 descriptors were not explained in Fig. 2.

3. Dicaffeoylquinic was identified as 4,5 by MS/MS but 3,5 by isolation.

4. Information could be shared in Supporting Information:

The list of 116 collected samples related to this plant may have been shared in Supporting information or online database.

The collected literature data on previously reported bioactivity (Reply 8) could also be added to Supporting information, taken that authors spent time to analyze the data (Note. cynaroside % Inh values must be mistaken 0.21 and 0.59).

The table with yields of extraction. (Reply 17)

The fact that vehicle control being activated by antigen in M&M (Reply 15)

HPLC chromatograms of isolated compounds (Reply 24).

The possible identity of major peak at 5.15-5.22 (could be mentioned in the main text).

5. Very nice and clear NMR analysis and literature comparison in Supporting Information. On which basis author confirmed the correct position of glucoside luteolin-7-O-glucoside when there is no HMBC correlation H1'' to C-7?

6. Semi-quantification is based on AUC of detected mass. Could relative amount per g of plant extract be calculated?

6. Some English errors could be corrected:

p.17 l.13, l.16 was/were (*MSI 1)

Supporting: diCaffeoyl, lutoelin-7-O-D-glucoside etc.

p.4 line 7 *considered as safe

p.11 l.18 *is possibly related

7. PLOS authors have the option to publish the peer review history of their article (what does this mean?). If published, this will include your full peer review and any attached files.

Reviewer #1: **Yes: **Michal Korinek

---

## [Author Response · Author response to Decision Letter 1]

24 Jan 2022

In the revised version, we have re-checked and corrected the reference list. References 28 and 33 are replaced with other relevant papers. DOI of reference 23 is corrected. The author’s names and journal abbreviations are completed and corrected. The titles are now displayed in sentence-format (capitalize the first letter of the first word). According to the reviewer’s comments, the experimental data including extraction yields, HPLC chromatograms of isolated compounds and the in-house library of compounds related to X.tridentata are added into supporting information. Typos and grammatical errors in the manuscript and supporting information are reviewed and corrected. In addition, we have updated the name of affiliation #6 since it has been recently changed (Jan 6, 2022). Please kindly find more detail about our responses to the reviewer in the “Response to Reviewers” file

---

## [Editor Report · Decision Letter 2]

18 Feb 2022

PONE-D-21-29347R2LC-MS/MS metabolomics-facilitated identification of the active compounds responsible for anti-allergic activity of the ethanol extract of Xenostegia tridentataPLOS ONE

Dear Dr. Chatwichien,

Thank you for submitting your manuscript to PLOS ONE. After careful consideration, we feel that it has merit but does not fully meet PLOS ONE’s publication criteria as it currently stands. Therefore, we invite you to submit a revised version of the manuscript that addresses the points raised during the review process.

The authors did not fully address Reviewer's and Academic Editor's concerns. Specifically, PC1 and PC2 descriptors should be given in the caption for Figure 2, regardless of their introduction in the main text. Figure captions should stand alone and be readable without referencing to the main text.

Regarding Reviewer's suggestion about expressing compound masses as relative amounts per gram, I am really convinced that the reviewer asked the authors to present such a calculation (as a table, presumably), not just to answer if this possibility exist.

Please do not randomly capitalize words (compounds particularly), even if abbreviation is provided in parenthesis (e.g., P5L6, P5L7, P10L21, P10L23, P18L10-11, and further in the text).

Citations in square brackets in the text should stand before a comma.

Please do not spell out "nanometers" (P7L15). Using "nm" only is quite acceptable. Also, using "AlCl_3_" without previous description is also perfectly fine (P7L14). Do not express centrifugation speed in rpm (P8L1). Use "*g*" instead (it needs computing), since equal rotation (rpm) in rotors with different diameters produce different g force. Therefore, *g* is a constant, while rpm is not. Do not use semicolons when listing items after colons, but use commas instead.

*Merremia tridentata* is a synonym of the studied plant species and cannot be referred to as "other plants in the Convolvulaceae family" (P17L4). Moreover, using vernacular expressions such as "other plants" (might present other individuals) instead of "plant species" is not scientifically justified.

We look forward to receiving your revised manuscript.

Kind regards,

Branislav T. Šiler, Ph.D.

Academic Editor

PLOS ONE
---

## [Author Response · Author response to Decision Letter 2]

2 Mar 2022

In the revised version, we have added the descriptions of PC1 and PC2 to the caption for Fig 2. The words that were randomly capitalized or unnecessary to be spelled out are corrected according to editor’s suggestions. The citations in square brackets are moved to stand before commas and periods. The word “in other plants” in P17L4 is replaced with the more appropriate word “plant species”. Fig 4 is revised to show the relative levels of the four significant metabolite features from the same ionization mode (ESI+ mode). The word “semi-quantitation” in main text is replaced with “relative metabolite level”, to avoid any misunderstanding that the AUC values could be directly used for quantification. The changes are highlighted in yellow in the “Revised Manuscript with Track Changes” file. In addition, please kindly find more detail about our responses in the “Response to Reviewers” file. 

Regarding the reviewer’s suggestion on the calculation of relative amount of compound per gram extract, we think that, based on the results that we currently have, we can not obtain the accurate values. Further quantitative analysis by using standard compounds will need to be performed (which is beyond our focus for this work). However, the putative values could be calculated based on assumptions. The calculation and result are shown in the “Response to Reviewers” file.

---

## [Editor Report · Decision Letter 3]

3 Mar 2022

LC-MS/MS metabolomics-facilitated identification of the active compounds responsible for anti-allergic activity of the ethanol extract of Xenostegia tridentata

PONE-D-21-29347R3

Dear Dr. Chatwichien,

We’re pleased to inform you that your manuscript has been judged scientifically suitable for publication and will be formally accepted for publication once it meets all outstanding technical requirements.

Kind regards,

Branislav T. Šiler, Ph.D.

Academic Editor

PLOS ONE
---

## [Editor Report · Acceptance letter]

23 Mar 2022

PONE-D-21-29347R3 

LC-MS/MS metabolomics-facilitated identification of the active compounds responsible for anti-allergic activity of the ethanol extract of *Xenostegia tridentata*

Dear Dr. Chatwichien:

I'm pleased to inform you that your manuscript has been deemed suitable for publication in PLOS ONE. Congratulations! Your manuscript is now with our production department. 

Kind regards, 

on behalf of

Dr. Branislav T. Šiler 

Academic Editor

PLOS ONE